# Prevalence of Germline Mutations in Cancer Predisposition Genes in Patients with Pancreatic Cancer or Suspected Related Hereditary Syndromes: Historical Prospective Analysis

**DOI:** 10.3390/cancers15061852

**Published:** 2023-03-20

**Authors:** Arianna Dal Buono, Laura Poliani, Luana Greco, Paolo Bianchi, Monica Barile, Valentina Giatti, Cristiana Bonifacio, Silvia Carrara, Alberto Malesci, Luigi Laghi

**Affiliations:** 1Department of Gastroenterology, IRCCS Humanitas Research Hospital, Rozzano, 20089 Milan, Italy; 2Gastroenterology and Endoscopy, IRCCS Ospedale San Raffaele and University Vita-Salute San Raffaele, 20132 Milan, Italy; 3Laboratory of Molecular Gastroenterology, Department of Gastroenterology, IRCCS Humanitas Research Hospital, Rozzano, 20089 Milan, Italy; 4Medical Analysis Laboratory, IRCCS Humanitas Research Hospital, Rozzano, 20089 Milan, Italy; 5Radiology Department, IRCCS Humanitas Research Hospital, Rozzano, 20089 Milan, Italy; 6Department of Medicine and Surgery, University of Parma, 43125 Parma, Italy

**Keywords:** pancreatic cancer, hereditary syndrome, pathogenic variant, germline mutation, cancer predisposition

## Abstract

**Simple Summary:**

Approximately 5–10% of all pancreatic adenocarcinomas (PDACs) are caused by highly penetrant pathogenic germline variants (PVs). Specific surveillance programs and eventual targeted oncological therapies can be offered to patients carrying some of the known PVs. We prospectively investigated the prevalence of germline PVs in cancer-predisposing genes in patients referred for genetic evaluation at our institution. In our cohort, 20.1% of the tested subjects harbored at least one PV in the genes of interest. Since the mutational burden in patients affected by PDAC or referred for a suspected related hereditary syndrome is high, the incorporation of genetic testing and the adoption of multiple-gene panels within the multidisciplinary management of this disease would be beneficial and desirable.

**Abstract:**

We investigate the prevalence of germline mutations in cancer predisposition genes in patients with pancreatic ductal adenocarcinoma (PDAC) or suspected related hereditary syndromes. Methods: we enrolled for NGS with an Illumina TrueSight Cancer panel comprising 19 CPGs and 113 consecutive subjects referred to cancer genetic clinics for metastatic PDAC, early onset PDAC, suspected hereditary syndrome, or positive family history. Results: Overall, 23 (20.1%) subjects were carriers of 24 pathogenetic variants (PVs). We found 9 variants in *BRCA2* (37.5%), 6 in *CDKN2A* (25%), 3 in *ATM* (12.5%), 2 in *BRCA1* (8.3%), 1 in *CHEK2* (4.1%), 1 in *PALB2* (4.1%), 1 in *MITF* (4.1%), and 1 in *FANCM* (4.1%). A double PV (*BRCA1* plus *BRCA2*) was found in 1 subject. We observed a nearly 30% (16/55) mutational rate in the subgroup of subjects tested for the suspected syndromes (PDAC and other synchronous or metachronous tumors or an indicative family history), and the frequency was significantly higher than that in patients with only metastatic PDAC (*p* = 0.05). In our cohort, 39 variants of unknown significance (VUS) were identified, most of which (16/39, 41%) in genes belonging to the Lynch syndrome spectrum. Conclusion: A clinically relevant proportion of pancreatic cancer is associated with mutations in known predisposition genes. Guidelines instructing on an adequate selection for accessing genetic testing are eagerly needed. The heterogeneity of mutations identified in this study reinforces the value of using a multiple-gene panel in pancreatic cancer.

## 1. Introduction

Pancreatic ductal adenocarcinoma (PDAC) is globally one of the most life-threatening malignant neoplasms, with a 5-year survival of around 5% and a 10-year survival of less than 1%, which have been almost unmodified over the past 30 years [1,2]. PDAC ranks 12th among the most common cancers all over the world, with a global incidence of approximately 450.000 cases [3,4].

PDAC research has shown growth in all areas, spanning early diagnosis, surgical techniques, and oncological therapy. An emerging research field is undoubtedly the investigation of possible genetic factors predisposing to PDAC given the estimates of 10% of patients with PDAC having a pathogenic germline alteration [5].

Familial pancreatic cancer (FPC) is suspected whenever two or more first-degree relatives are affected by PDAC [6]. For these patients and their families, global investigational surveillance programs offering annual abdominal magnetic resonance (MRI) are actively recruiting and ongoing in different countries, including Italy [7]. Beyond the clinical definition of FPC, approximately 5–10% of all PDAC are caused by highly penetrant germline pathogenic variants (PVs) [8,9,10] that usually result in distinguished hereditary cancer syndromes. Chiefly, heterozygous germline mutations in *BRCA1* and *BRCA2* predispose to the hereditary breast and ovarian cancer syndrome (HBOC), which carries an absolute risk of PDAC of 5–10% (i.e., four- and sevenfold greater than that of the general population) [11]. In addition to *BRCA* syndromes, Lynch syndrome (LS), familial atypical multiple-mole melanoma syndrome (FAMM), hereditary pancreatitis, Peutz–Jeghers syndrome, and Li–Fraumeni syndrome, which are caused by mutations in *MLH1*, *MSH2*, *MSH6*, and *PMS2,* mismatch repair genes (MMR), *PRSS1*, *STK11*, and *TP53,* respectively, confer an increased lifelong risk of PDAC varying from 5 to 15% [12,13] (Table 1). More recently, accumulating evidence has also associated *PALB2*, *ATM*, *BMPR1A*, and *SMAD4* with FPC [14,15,16] (Table 1), currently included in the suggested panel for genetic PDAC testing by the National Comprehensive Cancer Network (NCCN) guideline 2020 [17].

Identifying germline variants in susceptibility genes allows, on the one hand, for offering specific surveillance programs to individuals at high risk for the early detection of cancers besides PDAC (e.g., breast cancer in HBOC, colorectal cancer (CRC) in LS, or melanoma in FAMM). On the other hand, the genetic testing of affected patients allows for selecting them for poly (adenosine diphosphate-ribose) polymerase inhibitor (PARPi) and/or platinum-based treatment in the case of *BRCA1/2* pathogenic variants, for checkpoint inhibitor therapy in the case of MMR gene defects, or for additional developing therapies [18,19,20].

The current NCCN guideline [17] recommends that all patients affected by PDAC undergo germline genetic testing irrespectively of other factors (i.e., family history, age, synchronous/metachronous neoplasms). Oher guidelines for the genetic testing of patients with PDAC are rather restrictive, indicating the analysis of *BRCA 1/2* only in patients with metastatic neoplasia (for accessing second-line chemotherapy), and in subjects with a relevant family history (i.e., at least two first-degree relatives with PDAC or ≥three members diagnosed with PDAC) [21,22,23]. Testing for PVs in other genes is regulated by guidelines concerning organ-specific predisposing syndromes (i.e., CRC, breast cancer, gynecological cancer, and melanoma).

Methodologically, the use of multigene panels for genetic testing is becoming widespread, since selecting genes exclusively on the basis of clinical and family history may miss several PVs [5,17]. The recent development of next-generation DNA sequencing (NGS) technologies favors a growth in the use of such panels for the investigation of potentially involved genes in the development of oncological diseases.

We investigate the prevalence of germline pathogenic variants in cancer-predisposing genes in patients with PDAC or suspected related hereditary syndromes.

## 2. Materials and Methods

### 2.1. Patients

This is a historical single-center (Humanitas Research Hospital, Rozzano, Italy) prospective study of consecutive patients. Individuals who had undergone cancer genetic counseling for pancreatic cancer or suspected related hereditary syndromes from 2018 to 2021 were included. Inclusion criteria for genetic testing were: (1) diagnosis of metastatic PDAC, (2) diagnosis of early onset PDAC (before the age of 50), (3) suspected hereditary syndrome (i.e., PDAC and another synchronous or metachronous tumor, PDAC, an indicative family history of hereditary syndromes, the diagnosis of related neoplasms, and a positive family history for PDAC or consistent hereditary syndrome), or (4) healthy subjects with positive family history (at least two first-degree relatives affected or three relative affected by PDAC). All patients agreed to and signed the informed consent for genetic testing and its consultation for research purposes. Data were retrieved from the patients’ charts and diagnostic reports and collected into an anonymized database.

### 2.2. Next-Generation Sequencing

Blood samples were screened for germline variants by using the TruSight Cancer panel (Illumina) covering 19 cancer predisposition genes (Appendix A), and then ran on the Illumina MiSeq platform according to the manufacturer’s standard protocol (Illumina Inc, San Diego, CA, USA).

Genomic DNA was fragmented and processed with sequencing adaptors and indices via polymerase chain reaction (PCR). Once the sample libraries had been denatured into single-stranded DNA, they were hybridized into specific biotin-labeled probes for the targeted regions. By adding streptavidin beads able to bind to the biotinylated probes, the pool was further enriched. Afterwards, the streptavidin beads bound to biotinylated DNA fragments were pulled down, eluted from the beads, and hybridized for an enrichment reaction followed by PCR amplification. The targeted library was inserted onto the MiSeq platform for cluster generation and successive sequencing.

### 2.3. Bioinformatic Analyses and Variant Characterization

Quality and coverage data analyses were performed using onboard MiSeq Reporter software (Illumina Inc., San Diego, CA, USA), which is the mean sequencing coverage for the regions leveled by the Trusight panel of 366.2×. Across all samples, the fraction of the targeted regions with ≥30× coverage was around 94.6%, and 89.1% of all regions targeted by the Trusight panel had ≥100× coverage.

Thereafter, sequencing data were aligned using Burrows–Wheeler Aligner (BWA) software. Genetic variants were detected using GATK software and were exposed to further analysis both if the calculated genotype quality was ≥99 and if the site was identified as a heterozygous or homozygous variant site. Using ANNOVAR, Variant Interpreter (Illumina), and Sophia DDM (Sophia Genetics), the detected variants were subsequently annotated. Polymorphisms at >1% frequency were deleted using the Genome Aggregation Database (https://gnomad.broadinstitute.org/, accessed on 4 July 2022), 1000 Genomes (https://www.internationalgenome.org/home, accessed on 4 July 2022), and the Exome Sequencing Project (https://evs.gs.washington.edu/EVS/, accessed on 4 July 2022). Variants were classified according to the American College of Medical Genetics and Genomics/Association for Molecular Pathology criteria [24] into pathogenic (Class 5), likely pathogenic (Class 4), or of uncertain significance (VUS) (Class 3). The employed methodologies allow for detecting large insertions/deletions in the dry phase of the sequencing process. Class 4 and 5 or novel (not previously reported at the time of our analysis) variants were confirmed with Sanger sequencing.

In the employed panel, high-penetrance genes involve a lifetime risk of cancer >= 40%, while the risk associated with “moderate” genes is usually <40%, although this notion does not invariably apply to fully penetrant methylated CpG islands (CpGs) (e.g., *NF1*) [25]. We first retrieved NGS sequencing data for CpGs that were strongly associated with PDAC, and both PVs and VUS were annotated. Genetic counselling was offered to all family’s members of the patients diagnosed with a PV.

## 3. Results

Over the study period, a total of 113 individuals comprising 82 females (72.6%) and 31 males (27.4%) underwent genetic testing after counseling at our institution. In our cohort, 101/113 (89.4%) were affected by PDAC. Of these, 31 (27.4%) patients were tested because of metastatic PDAC, 15 (13.3%) for juvenile PDAC, and 55 (48.6%) subjects were tested for suspected hereditary syndrome. Within this category, the suspicion of hereditary syndrome depended on the history of synchronous or metachronous tumors (mostly breast cancer) for 26 (23%) subjects, and on family history for 28 (24.8%); 1 patient was affected by PDAC with microsatellite instability. Lastly, in 12 subjects (10.6%) the indication for genetic testing was motivated only by family history, suggestive of hereditary pancreatic cancer. Table 2 summarizes the clinical features of the included patients.

Overall, 23 (20.3%) subjects were carriers of 24 PVs (all class 5). We also identified 39 VUS in 35 subjects (31%). Lastly, 55 subjects (48.6%) did not harbor any variant in the evaluated genes (both PVs and VUS). Table 3 and Table 4 elucidate the details of the detected PVs. Specifically, we found 9 variants in *BRCA2* (37.5%), 6 in *CDKN2A* (25%), 3 in *ATM* (12.5%), 2 in *BRCA1* (8.3%), 1 in *CHEK2* (4.1%), 1 in *PALB2* (4.1%), 1 in *MITF* (4.1%) and 1 in *FANCM* (4.1%). The subject with double PVs (a patient with PDAC and previous double breast cancer) harbored PVs in both *BRCA1* and *BRCA2*.

The greatest number of identified VUS concerned genes belonging to the LS spectrum (16/39, 41%; *MLH1* 5/39 12.8%, *MSH2* 4/39, 10.3%, *MSH6* 4/39, 10.3% and *PMS2* 3/39, 7.7%), followed by *ATM* (9, 23.1%), *BRCA1/* 2 (6, 15.4%), *PALB2* (2, 5.1%), *APC* (2, 5.1%) and *CDKN2A*, *BMPR1A*, *BRIP1* and *CDK4* (1 each, 2.6%).

Concerning the indications for genetic testing, carriers of PVs were 3 patients with metastatic PDAC (3/31, 9.7%), 2 patients with juvenile PDAC (2/15, 13.3%), 16 patients tested because of any suspect of a hereditary syndrome (16/55, 29.1%) and 2 cases tested only for family history (2/12, 16.7%). Statistical analysis revealed a higher frequency of PVs in patients tested because of a suspected hereditary syndrome compared to those tested for metastatic PDAC (*p* = 0.06), and a significantly higher frequency of PVs in those with PDAC and a consistent family history (9/28, 32.1%; *p* = 0.05).

Table 2 reports the statistical analysis regarding PV rates with respect to the indication of the genetic test.

## 4. Discussion

In our study, we identified an overall PV rate of 20.3% in the selected subjects undergoing genetic testing for PDAC. This frequency was higher than that previously reported in the literature in unselected cohorts, in which it ranged from 5 to 15% [5,26,27,28,29,30]. The suspicion of a hereditary syndrome (for personal history of related synchronous or metachronous tumors, or for family history) mainly drove patient selection in our cohort; we also tested subjects affected by juvenile PDAC, a criterion that is not included in the Italian guidelines [20].

Our data show that the use of nonrestrictive guidelines allow for identifying a relevant proportion of carriers of a PVs associated with a hereditary syndrome; thus, the application of surveillance programs could reduce the overall burden of associated neoplasms [29]. We reported a high rate of PVs in subjects tested for the suspected inherited syndromes, approaching one-third of the patients. In detail, the main identified PVs were in *BRCA1/2* (47.8%) and *CDKN2A* (6.2%). Pathogenic *BRCA1/2* mutations confer an increased risk of developing neoplasms, especially of the breasts and ovaries: in our series, three patients with these PVs showed a typical phenotype, highly consistent with the syndrome (including the patient with double mutation). A congruous genotype–phenotype correlation was also observed for patients carrying a PV in *CDKN2A*.

In our series, the main genes presenting a VUS were those of MMR (16/39, 41%), highlighting the relevance of data maintenance for updating the encountered variants.

We observed a higher rate of PVs in the tested categories of patients because of suspected hereditary syndromes as compared to patients tested merely because of metastatic PDAC, which is maximized by a positive family history (3/31 vs. 16/55; 9.7% vs. 29.1%). This result underlines that the proper selection of the patients for genetic testing can improve the performance of the test while containing costs.

As screening for PDAC is unfeasible due to its low prevalence in the general population, the surveillance of high-risk groups is suggested as the approach for increasing the probability of an early detection. Such an approach, referred to as “define–enrich–find,” (DEF), concerns patients with intraductal papillary mucinous neoplasms (i.e., IPMNs), with new onset diabetes (i.e., NOD), chronic pancreatitis, and those with familial pancreatic cancer or predisposing inherited conditions [31]. In this context, multigene testing is gaining widespread use across different countries and medical systems [32,33,34], a plan that also envisiond enrollment through websites and sample collection from home [35].

There is a consensus as to the management of patients with inherited predispositions increasing the risk of PDAC, who should undergo yearly surveillance by endoscopic ultrasound (EUS) or magnetic resonance (MR) [36]. The precise extent to which such an approach would be beneficial over time remains to be ascertained, although the surveillance of *CDNK2A* mutation carriers was relatively successful [37]. In a Dutch surveillance study of high-risk mutation carriers, the diagnostic yield of PDAC was substantial, although the timely identification of resectable lesions was challenging, indicating that imaging does not deliver optimal results, and more sensitive diagnostic approaches such as biomarkers are needed [38].

A recent metanalysis indicates that, in high-risk individuals under surveillance, high-risk abnormalities (at MR or EUS plus fine needle aspiration) were significantly associated with surgical appropriateness (intended as the resection of premalignant and malignant lesions) and beneficial in 4 out of 10 cases [39]. In accordance with these data, in our clinical practice, surveillance with MR or EUS is advised for carriers of PVs for PDAC.

Our data show that PDAC predispositions play a leading role within hereditary syndromes. The implementation and clarification of the criteria for accessing the genetic testing in patients with PDAC and their families is warranted. In certain European contexts, such as in Italy, although the guideline remains restrictive (i.e., only the analysis of the PVs of *BRCA1/2* is foreseen) [21], scientific societies, such as the Italian Association for the Study of the Pancreas (AISP), promoted a registry of subjects at increased risk of PDAC, starting a parallel prevention program [40]. The access criteria of this registry, in line with international recommendations [40], include the presence of PVs in several genes (*BRCA1/2*, *CDKN2A*, *PALB1*, *STK11*, and MMR, and, recently, *ATM*) [40,41,42].

In conclusion, our data support the need for further studies to improve our understanding of the impact of genetics on the development of PDAC, which is a main player among tumors in which hereditary syndromes are implicated. Through this process, a broader identification of subjects affected by a hereditary syndrome might be possible, together with their access to dedicated surveillance programs for cancer prevention and an eventual possible improvement in target therapies.

## 5. Conclusions

A relevant proportion of PDAC is associated with mutations in known predisposition genes. Guidelines instructing on the adequate selection for accessing genetic testing are eagerly needed. The heterogeneity of mutations identified in this study reflects the value of using a multiple-gene panel in pancreatic cancer.

## Figures and Tables

**Table 1 cancers-15-01852-t001:** Hereditary syndromes predisposing to pancreatic ductal adenocarcinoma [15,17].

Syndrome	Gene	Cancer Type and Risk
Breast and ovarian cancer syndrome (HBOC)	*BRCA1/2*	PDAC: relative risk of 2–10%, lifelong risk 3–10%. High risk of breast, ovarian, and prostate cancer.
Lynch syndrome (LS)	*MLH1* *MSH2* *MSH6* *PMS2* *EPCAM*	PDAC: for MLH1 relative risk of ~7%, lifelong risk ~6%. Colorectal, gastric, and endometrial cancer. Phenotype depends on the specific gene and mutation.
Familial atypical multiple mole melanoma syndrome (FAMM)	*CDKN2A*	PDAC: relative risk of 13–39%, lifelong risk ~17%. High risk of malignant melanoma.
Peutz–Jeghers syndrome	*STK11*	PDAC: relative risk of 70–75%, lifelong risk > 25%. Gastrointestinal polyposis.
Li–Fraumeni syndrome	*TP53*	PDAC: ~7% of lifelong risk. High risk of hematopoietic malignancies, breast cancer, central nervous system tumors, osteosarcomas, and soft-tissue sarcomas.
Others	*PALB2* *ATM* *BMPR1A SMAD4*	PDAC: relative risk ~2.5%, lifelong risk ~5%. Female breast and ovarian cancer.

PDAC: pancreatic ductal adenocarcinoma.

**Table 2 cancers-15-01852-t002:** Clinical features of the included patients and statistical analysis using Fisher’s exact test of mutational rates.

Patients’ Features	*n* = 113 *n*, (%)	PVs Carrier *n* (%)	*p*-Value
Sex (female)	82 (72.6)		
Indication for genetic testing (1) Metastatic PDAC (2) Juvenile PDAC § (3) Suspected syndrome PDAC and S/M related tumor † PDAC and family history PDAC MSI (4) Family history	31 (27.4) 15 (13.3) 55 (48.6) 26 (23.0) 28 (24.8) 1 (0.9) 12 (10.6)	3/31 (9.7) * 2/15 (13.3) # 16/55 (29.1) 7/26 (26.9) ç 9/28 (32.1) ° 0/1 (0.0) 2/12 (16.7) &	0.06 ∇ 0.05 ∇

Table 2 summarizes the indications for genetic testing in our cohort: (1) the diagnosis of metastatic PDAC, (2) diagnosis of early onset PDAC (before the age of 50), (3) suspected hereditary syndrome (i.e., PDAC and another synchronous or metachronous related tumor, PDAC and indicative family history of hereditary syndrome, diagnosis of other related neoplasms and positive family history for PDAC or consistent hereditary syndrome), or (4) healthy subjects with at least two first-degree relatives affected or three relative affected by PDAC. ∇ Statistical analysis using Fisher’s exact test of mutational rates in patients with suspected syndrome and in those affected by PDAC with family history as compared to the group of patients with metastatic PDAC. § Juvenile: <50 years of age; * *BRCA2, ATM, PALB2*; # *BRCA2, BRCA1*; Ç *MITF*, *BRCA2* (*n* = 2), *CDKN2A, ATM, FANCM*, and double mutation *BRCA2* plus *BRCA1*; ° *CDKN2A* (*n* = 4), *BRCA2* (*n* = 3), *ATM, CHEK2*; & *BRCA2, CDKN2A*; PDAC: Pancreatic ductal adenocarcinoma; S/M: synchronous or metachronous; MSI: microsatellite instability † associated synchronous or metachronous tumors.

**Table 3 cancers-15-01852-t003:** Genetic variants in the study population.

Genetic Variant Details	*n* = 113 *n* (%)
Total mutations PVs (class 5) VUS	58 (51.3) 23 (20.1) 35 (31.0)
PVs * BRCA2* * CDKN2A* *ATM* *BRCA1* Others *	*n* = 24 9 (37.5) 6 (25.0) 3 (12.5) 2 (8.3)
VUS Lynch genes † *ATM* *BRCA ½* *PALB2* *APC* Others ‡	*n* = 39 16 (41.0) 9 (23.1) 6 (15.4) 2 (5.1) 2 (5.1) 4 (10.2)

* *PALB2*, *CHEK2*, *MITF* and *FANCM*. † *MLH1* (5/39, 12.8%), *MSH2* (4/39, 10.3%), *MSH6* (4/39, 10.3%), *PMS2* (3/39, 7.7%); ‡ *CDKN2A*, *BMPR1A*, *BRIP1* and *CDK4*. PVs: pathogenetic variants; VUS: variants of unknown significance.

**Table 4 cancers-15-01852-t004:** Pathogenic variants identified in our study.

Patient	Indication for Genetic Testing	Clinical Features	Gene	Nomenclature	Protein Change	Variant Interpretation
1	Ssyn	Unaffected; two first-degree relatives affected by melanomas and two first-degree relatives affected by PDAC	*CDKN2A*	c.142C>A	p.Pro48Thr	Class 5
2	Ssyn	Affected by metastatic PDAC and metachronous breast cancer	*MITF*	c.1255G>A	p.Glu419Lys	Class 5
3	FH	Unaffected; two first-degree and one second-degree relatives affected by PDAC	*BRCA2*	c.6469C>T	p.Gln2157Ter	Class 5
4	Ssyn	Affected by PDAC and metachronous melanoma, one first-degree relative affected by PDAC	*CDKN2A*	c.71G>C	p.Arg24Pro	Class 5
5	Ssyn	Affected by PDAC and metachronous melanoma, one first-degree relative affected by PDAC	*CDKN2A*	c.(?_-1)_(*1_?)del	p.0?	Class 5
6	Ssyn	Affected by PDAC with ≥2 first-degree relatives affected by PDAC	*BRCA2*	c.4229dupC	p.Ala1411Cysfs*3	Class 5
7	Ssyn	Affected by PDAC with ≥2 first-degree relatives affected by breast cancer	*BRCA2*	c.5073dupA	p.Trp1692MetfsTer3	Class 5
8	FH	Unaffected; two first-degree and one second-degree relatives affected by PDAC	*CDKN2A*	c.377T>A	p.Val126Asp	Class 5
9	Ssyn	Affected by breast cancer with ≥2 first-degree relatives affected by PDAC	*ATM*	c.217_218delGA	p.Glu73MetfsTer26	Class 5
10	Ssyn	Affected by breast cancer with ≥2 first-degree relatives affected by PDAC	*CHEK2*	c.660delA	p.Gly221GlufsTer6	Class 5
11	Ssyn	Affected by PDAC and metachronous breast cancer	*BRCA2*	c.1238delT	p.Leu413Hisfs*17	Class 5
12	jPDAC	Affected by PDAC before 50 years of age	*BRCA2*	c.3744_3747delTGAG	p.Ser1248Argfs*10	Class 5
13	Ssyn	Affected by PDAC with ≥2 first-degree relatives affected by breast cancer or PDAC	*CDKN2A*	c.301G>T	p.Gly101Trp	Class 5
14	mPDAC	Affected by metastatic PDAC	*BRCA2*	c.3028A>T	p.Arg1010Ter	Class 5
15	Ssyn	Affected by PDAC and metachronous breast cancer	*BRCA2*	c.2094delA	p.Gln699SerfsTer31	Class 5
16	mPDAC	Affected by metastatic PDAC with one first-degree relative affected by breast cancer (bilateral)	*ATM*	c.8147T>C	p.Val2716Ala	Class 5
17	Ssyn	Affected by PDAC and metachronous melanoma, one first-degree relative affected by melanoma	*CDKN2A*	c.301G>T	p.Gly101Trp	Class 5
18	Ssyn	Affected by breast cancer with ≥2 first-degree relatives affected by breast cancer or PDAC	*BRCA2*	c.7060C>T	p.Gln2354*	Class 5
19	Ssyn	Affected by PDAC and metachronous melanoma and breast cancer	*ATM*	c.5592delA	p.His1865MetfsTer52	Class 5
20	Ssyn	Affected by PDAC and metachronous breast cancer	*FANCM*	c.5101C>T	p.Gln1701Ter	Class 5
21	jPDAC	Affected by PDAC before 50 years of age	*BRCA1*	c.3756_3759delGTCT	p.Ser1253ArgfsTer10	Class 5
22	Ssyn	Affected by PDAC and metachronous bilateral breast cancer	*BRCA1* *BRCA2*	c.181T>G c.755_758del	p.Cys61Gly p.Asp252ValfsTer24	Class 5 Class 5
23	mPDAC	Affected by metastatic PDAC with one second-degree relative affected by breast cancer	*PALB2*	c.1266del	p.Val423Ter	Class 5

PDAC: pancreatic ductal adenocarcinoma, mPDAC: metastatic pancreatic ductal adenocarcinoma, jPDAC: juvenile <50 years of age at diagnosis, Ssyn: suspected syndrome (PDAC and S/M related tumor, PDAC and family history, and PDAC MSI), FH: family history.

## Data Availability

Data are available from the corresponding author.

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
