# Peer review of "Prevalence of Germline Mutations in Cancer Predisposition Genes in Patients with Pancreatic Cancer or Suspected Related Hereditary Syndromes: Historical Prospective Analysis"

_cancers, 2023, doi:10.3390/cancers15061852_

Round 1
Reviewer 1 Report (New Reviewer)
The manuscript "Prevalence of germline mutations in cancer predisposition genes in patients with pancreatic cancer or suspected related hereditarysyndromes: an historical prospective analysis" addresses the prevalence of germline mutations in cancer predisposition genes in a cohort of 113 patients enrolled according to indications that slightly extend the existing guidelines. The authors report a relatively high prevalence of these mutations indicating that panel sequencing can help clinical decision making.
The study is robust, sufficiently powered, well performed and the results are clearly reported and interpreted. As stated by referee #2 clinical information on the patients is insufficient. In particular, the question of whether patients carrying these mutations show specific clinical features should be addressed. The specific mutations should be indicated, eventually as a supplementary table. A discussion regarding the functional consequence of the mutations identified and their oncogenic potential must be added.
Author Response
Response to the editor(s) of ‘Cancers’
Dear Editor(s),
We sincerely thank you for giving us the opportunity to consider for potential publication a revised version of our original manuscript entitled “Prevalence of germline mutations in cancer predisposition genes in patients with pancreatic cancer or suspected related hereditary syndromes: an historical prospective analysis" by Dal Buono A et al.
We kindly thank the Reviewers for the precious comments. We are pleased to know that you appreciated the topic of our manuscript. The manuscript has been significantly revised and improved according to the received suggestions. Included below you can find a point-by-point response to the remarks.
Sincerely,
Prof Luigi Laghi
luigi.laghi@humanitas.it
luigiandreagiuseppe.laghi@unipr.it
Reviewer#1
The manuscript "Prevalence of germline mutations in cancer predisposition genes in patients with pancreatic cancer or suspected related hereditary syndromes: an historical prospective analysis" addresses the prevalence of germline mutations in cancer predisposition genes in a cohort of 113 patients enrolled according to indications that slightly extend the existing guidelines. The authors report a relatively high prevalence of these mutations indicating that panel sequencing can help clinical decision making.
The study is robust, sufficiently powered, well performed and the results are clearly reported and interpreted. As stated by referee #2 clinical information on the patients is insufficient. In particular, the question of whether patients carrying these mutations show specific clinical features should be addressed. The specific mutations should be indicated, eventually as a supplementary table. A discussion regarding the functional consequence of the mutations identified and their oncogenic potential must be added.
RE: Thanks for Your comment. We are glad to hear that You appreciated the revised version of our paper. We have improved Table S2 that has become Table 4, according to Your suggestions.
Reviewer 2 Report (New Reviewer)
In this manuscript, the authors provide an analysis of germline mutations and VUS tested by the TSO500 panel in blood samples obtained from a heterogeneous collective of 113 individuals undergoing genetic counselling for pancreatic cancer (PDAC) predisposition. 20% of individuals were found to bear a pathogenic variant and mostly belonged to the subgroup of patients with PDAC and positive family history.
The manuscript has been previously reviewed and I have been granted access both to the first and to the revised submission. The topic of the manuscript is interesting, but there is still room for improvement:
1- The different subgroups of individuals should be better and consistently defined. Paragraph 2.1 -Patients: group 3 is heterogenous and the different individuals included here should be better specified. A table with clear criteria should be provided and also used to report the results. Tis is now done in Table 2, which is confusing for the reader: e.g. the definition “PDAC and S/M related tumor appears both in subgroup 3 and in subgroup 4; symbols are not all explained in the legend.
2- Tables 2 and 4 could be combined in a single one, which should be better structured and explained, as also outlined in point 1 above.
3- The authors should introduce a paragraph related to the possible methodological limitations of the study: is the panel used able to detect large genomic rearrangements, which are know to occur in the BRCA genes?
4- The authors should be critically discuss the clinical relevance of their results in light of the missing screening programs for PDAC and the lack of therapeutic strategies for individual bearing a genetic risk for PDAC. At the end of paragraph 2.3 they affirm that “Genetic counselling was offered to all family’s members of the patients diagnosed with a PV”. What is the content of this counselling? What is exactly going to happen to these people? A cost-benefit analysis of their approach should be provided and the study should be put in a larger, international context.
Author Response
Response to the editor(s) of ‘Cancers’
Dear Editor(s),
We sincerely thank you for giving us the opportunity to consider for potential publication a revised version of our original manuscript entitled “Prevalence of germline mutations in cancer predisposition genes in patients with pancreatic cancer or suspected related hereditary syndromes: an historical prospective analysis" by Dal Buono A et al.
We kindly thank the Reviewers for the precious comments. We are pleased to know that you appreciated the topic of our manuscript. The manuscript has been significantly revised and improved according to the received suggestions. Included below you can find a point-by-point response to the remarks.
Sincerely,
Prof Luigi Laghi
luigi.laghi@humanitas.it
luigiandreagiuseppe.laghi@unipr.it
Reviewer#2
In this manuscript, the authors provide an analysis of germline mutations and VUS tested by the TSO500 panel in blood samples obtained from a heterogeneous collective of 113 individuals undergoing genetic counselling for pancreatic cancer (PDAC) predisposition. 20% of individuals were found to bear a pathogenic variant and mostly belonged to the subgroup of patients with PDAC and positive family history.
The manuscript has been previously reviewed and I have been granted access both to the first and to the revised submission. The topic of the manuscript is interesting, but there is still room for improvement:
RE: Thanks for Your comment. Included below you can find a point-by-point response to the remarks.
- The different subgroups of individuals should be better and consistently defined. Paragraph 2.1 -Patients: group 3 is heterogenous and the different individuals included here should be better specified. A table with clear criteria should be provided and also used to report the results. Tis is now done in Table 2, which is confusing for the reader: e.g. the definition “PDAC and S/M related tumor appears both in subgroup 3 and in subgroup 4; symbols are not all explained in the legend.
RE: Thanks for Your comment. We have improved Table 2 according to Your suggestions, clarifying the criteria for genetic testing, and improving the legend. We agree that the group 3 might appear heterogeneous, but we now provide detailed data in table 4, covering individual cases with class pathogenic variants.
- Tables 2 and 4 could be combined in a single one, which should be better structured and explained, as also outlined in point 1 above.
RE: Thanks for Your comment. We have accordingly restructured the tables in the current version.
- The authors should introduce a paragraph related to the possible methodological limitations of the study: is the panel used able to detect large genomic rearrangements, which are know to occur in the BRCA genes?
RE: Thank You for Your comment. Methodologically our sequencing technique allows to cover large deletions and duplications and copy number variations are accurately detected. This has been specified in the text accordingly.
- The authors should be critically discuss the clinical relevance of their results in light of the missing screening programs for PDAC and the lack of therapeutic strategies for individual bearing a genetic risk for PDAC. At the end of paragraph 2.3 they affirm that “Genetic counselling was offered to all family’s members of the patients diagnosed with a PV”. What is the content of this counselling? What is exactly going to happen to these people? A cost-benefit analysis of their approach should be provided and the study should be put in a larger, international context.
RE: Thanks for Your comment. We have improved the discussion section as suggested in several points.
Reviewer 3 Report (New Reviewer)
I read the revised manuscript and the point by point response to reviewer's comments. I think that the Authors adequately replied to the suggestions and the paper is suitable for publication.
Author Response
Response to the editor(s) of ‘Cancers’
Dear Editor(s),
We sincerely thank you for giving us the opportunity to consider for potential publication a revised version of our original manuscript entitled “Prevalence of germline mutations in cancer predisposition genes in patients with pancreatic cancer or suspected related hereditary syndromes: an historical prospective analysis" by Dal Buono A et al.
We kindly thank the Reviewers for the precious comments. We are pleased to know that you appreciated the topic of our manuscript. The manuscript has been significantly revised and improved according to the received suggestions. Included below you can find a point-by-point response to the remarks.
Sincerely,
Prof Luigi Laghi
luigi.laghi@humanitas.it
luigiandreagiuseppe.laghi@unipr.it
Reviewer#3
I read the revised manuscript and the point by point response to reviewer's comments. I think that the Authors adequately replied to the suggestions and the paper is suitable for publication.
RE: Thanks for Your comment. We are glad to hear that You appreciated the revised version of our paper.
Reviewer 4 Report (New Reviewer)
I feel that the results are not correlated with clinical presentation. If the authors want to use the variants to determine risk of PDAC, the results need to be validated in larger sets. At this point, I feel that publishing these results is premature. I would advise the resubmission with more convincing data.
Author Response
Response to the editor(s) of ‘Cancers’
Dear Editor(s),
We sincerely thank you for giving us the opportunity to consider for potential publication a revised version of our original manuscript entitled “Prevalence of germline mutations in cancer predisposition genes in patients with pancreatic cancer or suspected related hereditary syndromes: an historical prospective analysis" by Dal Buono A et al.
We kindly thank the Reviewers for the precious comments. We are pleased to know that you appreciated the topic of our manuscript. The manuscript has been significantly revised and improved according to the received suggestions. Included below you can find a point-by-point response to the remarks.
Sincerely,
Prof Luigi Laghi
luigi.laghi@humanitas.it
luigiandreagiuseppe.laghi@unipr.it
Reviewer#4
I feel that the results are not correlated with clinical presentation. If the authors want to use the variants to determine risk of PDAC, the results need to be validated in larger sets. At this point, I feel that publishing these results is premature. I would advise the resubmission with more convincing data.
RE: Our study presents a cohort numerically consistent with those recently published, please see
- Rapposelli IG, Zampiga V, Cangini I, eta al. Comprehensive analysis of DNA damage repair genes reveals pathogenic variants beyond BRCA and suggests the need for extensive genetic testing in pancreatic cancer. BMC Cancer. 2021 May 26;21(1):611
- Uson PLS Jr, Samadder NJ, Riegert-Johnson D, et al. Clinical Impact of Pathogenic Germline Variants in Pancreatic Cancer: Results From a Multicenter, Prospective, Universal Genetic Testing Study. Clin Transl Gastroenterol. 2021 Oct 8;12(10):e00414
- Schwartz M, Korenbaum C, Benfoda M, et al. Familial pancreatic adenocarcinoma: A retrospective analysis of germline genetic testing in a French multicentre cohort. Clin Genet. 2019 Dec;96(6):579-584)
Accordingly, we believe that it is not premature at all to grant access to publication to our data as well.
Round 2
Reviewer 1 Report (New Reviewer)
The authors have adequately addressed the issues raised by the referees.
This manuscript is a resubmission of an earlier submission. The following is a list of the peer review reports and author responses from that submission.
Round 1
Reviewer 1 Report
Poliani et al. reported a nice study on the prevalence of germline mutations in cancer predisposition genes in PDAC or suspected related hereditary syndrome patients. The study is very well executed and I do not have any major concern.
I just have a minor comment about some missing citations that performed same kind of work related to PDAC: such as Grant et al., 2014, Chaffee et al., 2017, Erin et al., 2015...
Author Response
Response to the editor(s) of ‘Cancers’
Dear Editor(s),
We sincerely thank you for giving us the opportunity to consider for potential publication a revised version of our original manuscript entitled “Prevalence of germline mutations in cancer predisposition genes in patients with pancreatic cancer or suspected related hereditary syndromes: a
retrospective analysis " by Poliani et al.
We kindly thank the Reviewers for the precious comments. We are pleased to know that you appreciated the topic of our manuscript. The manuscript has been significantly revised and improved according to the received suggestions. Included below you can find a point-by-point response to the remarks.
Sincerely,
Prof Luigi Laghi
luigi.laghi@humanitas.it
Reviewer#1
Poliani et al. reported a nice study on the prevalence of germline mutations in cancer predisposition genes in PDAC or suspected related hereditary syndrome patients. The study is very well executed, and I do not have any major concern.
Re: Thank You for Your comment. We are glad to know You appreciated our paper.
I just have a minor comment about some missing citations that performed same kind of work related to PDAC: such as Grant et al., 2014, Chaffee et al., 2017, Erin et al., 2015...
Re: Thank You for Your comment. We have added the suggested citations in the discussion accordingly.
Reviewer 2 Report
In this manuscript, the authors investigated the germline variants of 19 CPGs from TruSight Cancer panel (Illumina) on 113 patients for PDAC or suspected PDAC and other diseases with family history. The goal of the study is clinically important, but there lacks the systematic and clinically relevant analysis and conclusion of the data is vague and less significant. The data was not presented explicitly, which variants are tested of the 19 genes, what are their findings? Is it rather very likely that the 19 CPGs are PVs for pancancer, but where is the evidence to support the curation in pancreatic cancer?
minor
1. Line 24-5, “Most of the 24 variants (16/55, 29.1%) were detected in subjects tested for…”, this sentence is confusing, apparently, it refers to 16 out of 55 patients not variants, please rephrase.
2. The description of patient grouping is not clear. Line 156, what is “Class 5”, meaning all 5 classes? That did not match with table 1. What is the justification of grouping criteria? Why are other cancers than PDAC is included? Please modify table 1, be more specific with class categories, including age and sex for each.
3. Line 155-6, what is the difference between the term of “pathogenetic mutation” and “pathogenetic variants (PVs)”. Suppl. Table to list the identified PVs, VUS, etc in each class.
Author Response
Response to the editor(s) of ‘Cancers’
Dear Editor(s),
We sincerely thank you for giving us the opportunity to consider for potential publication a revised version of our original manuscript entitled “Prevalence of germline mutations in cancer predisposition genes in patients with pancreatic cancer or suspected related hereditary syndromes: a
retrospective analysis " by Poliani et al.
We kindly thank the Reviewers for the precious comments. We are pleased to know that you appreciated the topic of our manuscript. The manuscript has been significantly revised and improved according to the received suggestions. Included below you can find a point-by-point response to the remarks.
Sincerely,
Prof Luigi Laghi
luigi.laghi@humanitas.it
Reviewer#2
In this manuscript, the authors investigated the germline variants of 19 CPGs from TruSight Cancer panel (Illumina) on 113 patients for PDAC or suspected PDAC and other diseases with family history. The goal of the study is clinically important, but there lacks the systematic and clinically relevant analysis and conclusion of the data is vague and less significant. The data was not presented explicitly, which variants are tested of the 19 genes, what are their findings? Is it rather very likely that the 19 CPGs are PVs for pancancer, but where is the evidence to support the curation in pancreatic cancer?
Re: Thank You for Your comment. The NGS technique is a sequencing method, that is to say that no specific variants were searched but that the test resulted in individuating all the different variants possible in our patients. Our study is not focused on the cure, rather on the prevalence and on patients’ features associated with a higher rate of PVs.
We have improved the text according to Your suggestions. Included below you can find a point-by-point response to the remarks.
Minor
- Line 24-5, “Most of the 24 variants (16/55, 29.1%) were detected in subjects tested for…”, this sentence is confusing, apparently, it refers to 16 out of 55 patients not variants, please rephrase.
Re: Thank You for Your comment. We rephrased accordingly.
- The description of patient grouping is not clear. Line 156, what is “Class 5”, meaning all 5 classes? That did not match with table 1. What is the justification of grouping criteria? Why are other cancers than PDAC is included? Please modify table 1, be more specific with class categories, including age and sex for each.
Re: Thank You for Your comment. We need to clarify that in our text “class” specifically and uniquely refers to the genetic variants’ classification (see. according to the American College of Medical Genetics and Genomics/Association for Molecular Pathology criteria, and https://www.ncbi.nlm.nih.gov/clinvar/). We never used the term “class” with regard to patients’ subgroups.
We have numbered the categories of patients both in the Methods section and in Table 1 to make them clearer, as You suggested. Distinguishing age and sex within these groups is not relevant for the study’s aim.
- Line 155-6, what is the difference between the term of “pathogenetic mutation” and “pathogenetic variants (PVs)”. Suppl. Table to list the identified PVs, VUS, etc in each class.
Re: Thank You for Your comment. We corrected the text accordingly. Moreover Supplementary Table S2 has been added to the paper to give details on the PVs and VUS, as You suggested.
Reviewer 3 Report
Polaiani et al. took the very important topic of genetic predisposition to pancreatic cancer. This is a topic of high clinical relevance. Some patients harboring inherited germline predisposition to cancer may require dedicated clinical management, which will contribute to effective cancer treatment and the prevention of potential severe toxicities. Moreover, the knowledge about underlying genetic predisposition may lead to appropriate screening, management of comorbidities, as well as genetic counseling. Nevertheless, there are a few issues that should be addressed before publication:
Major:
- Authors stated that: "PDAC is expected to become the second 41 leading cause of cancer death in the United States by 2030 (3)". As we have 2022, these data should be related to facts from 2020, not predictions from 2014.
- "given the estimates of 10% of patients with PDAC have a pathogenic germline alteration (4)" - it is estimated that approximately 10% of malignancies occur in carriers of germline mutations predisposing to cancer. I think that the authors should show the percentage of PDAC in this context.
- The syndromes listed in lines 53-59 should be collected in a table with a proper description of cancer spectrum/risk and clinical comments.
- Line 60-62: What with pancreatic cancer? Do we have any program for these patients?
- line 67: Could you clarify if this recommendation was introduced in 2022? Of which genes? It would be more specified...
- My main concern is... how did you prepare the targeted list of genes? I am concerned about the lack of the PRSS1 gene.
- Did you look at the familial segregation of this variant? Did you provide the opportunity for genetic counseling for the patient's family?
- Found pathogenic variants should be listed in the Supplement. The whole data should be put in a repository.
Minor:
- "given the low sensitivity of the clinical picture in predicting the real risk of hereditary predisposition" - I am not sure if I properly understand this sentence. Could you rephrase it for me?
- genes listed in full text and supplementary materials should be written in Italica
- the abbreviation LS was introduced in the abstract, but it was not used in the next part of the abstract
- Table 1: missing % in first 4 rows
- Table 3 - p-value: what was compared? It is not clear (without description from full text) and should be added in the footer
I would like to congratulate you on your interesting paper. I will be glad to review it one more time after revision.
Best regards
Author Response
Response to the editor(s) of ‘Cancers’
Dear Editor(s),
We sincerely thank you for giving us the opportunity to consider for potential publication a revised version of our original manuscript entitled “Prevalence of germline mutations in cancer predisposition genes in patients with pancreatic cancer or suspected related hereditary syndromes: a
retrospective analysis " by Poliani et al.
We kindly thank the Reviewers for the precious comments. We are pleased to know that you appreciated the topic of our manuscript. The manuscript has been significantly revised and improved according to the received suggestions. Included below you can find a point-by-point response to the remarks.
Sincerely,
Prof Luigi Laghi
luigi.laghi@humanitas.it
Reviewer#3
Poliani et al. took the very important topic of genetic predisposition to pancreatic cancer. This is a topic of high clinical relevance. Some patients harboring inherited germline predisposition to cancer may require dedicated clinical management, which will contribute to effective cancer treatment and the prevention of potential severe toxicities. Moreover, the knowledge about underlying genetic predisposition may lead to appropriate screening, management of comorbidities, as well as genetic counseling. Nevertheless, there are a few issues that should be addressed before publication:
Re: Thank You for Your comment. We have improved the text according to Your suggestions. Included below you can find a point-by-point response to the remarks.
Major:
- Authors stated that: "PDAC is expected to become the second leading cause of cancer death in the United States by 2030 (3)". As we have 2022, these data should be related to facts from 2020, not predictions from 2014.
Re: Thank You for Your comment. We corrected accordingly.
- "given the estimates of 10% of patients with PDAC have a pathogenic germline alteration (4)" - it is estimated that approximately 10% of malignancies occur in carriers of germline mutations predisposing to cancer. I think that the authors should show the percentage of PDAC in this context.
Re: Thank You for Your comment. We added one ref with relevance to the issue, reporting to a prevalence close to 15% of PDAC patients from in silico analysis hunting for germline PVs.
- The syndromes listed in lines 53-59 should be collected in a table with a proper description of cancer spectrum/risk and clinical comments.
Re: Thank You for Your comment. We have added Table 1 accordingly.
- Line 60-62: What with pancreatic cancer? Do we have any program for these patients?
Re: Thank You for Your comment. We have added details on available programs, as suggested.
- line 67: Could you clarify if this recommendation was introduced in 2022? Of which genes? It would be more specified...
Re: Thank You for Your comment. We have added further details accordingly.
- My main concern is... how did you prepare the targeted list of genes? I am concerned about the lack of the PRSS1 gene.
Re: Thank You for Your comment. We agree with You that chronic pancreatitis in PRSS1 mutated patients is a condition at very high risk for PDAC. At our Institution we adopt a multi-gene panel (Illumina) specifically oriented to cancer predisposing genes that doesn’t include PRSS1. In case of clinical suspicion or radiological findings of chronic/calcific pancreatitis the patient is then tested for additionally for PRSS1/SPINK1.
- Did you look at the familial segregation of this variant? Did you provide the opportunity for genetic counseling for the patient's family?
Re: Thank You for Your comment. We have offered the opportunity for genetic counseling for the patients’ families and added a comment on that in the method section, as suggested.
- Found pathogenic variants should be listed in the Supplement. The whole data should be put in a repository.
Re: Thank You for Your comment. We have added a Supplementary Table S2 elucidating all the pathogenic variants found in our study population, as You suggested.
Minor:
- "given the low sensitivity of the clinical picture in predicting the real risk of hereditary predisposition" - I am not sure if I properly understand this sentence. Could you rephrase it for me?
Re: Thank You for Your comment. We improved the text accordingly.
- genes listed in full text and supplementary materials should be written in Italica
Re: Thank You for Your comment. We improved the text accordingly.
- the abbreviation LS was introduced in the abstract, but it was not used in the next part of the abstract
Re: Thank You for Your comment. We improved the text accordingly.
- Table 1: missing % in first 4 rows
Re: Thank You for Your comment. We improved the text accordingly.
- Table 3 - p-value: what was compared? It is not clear (without description from full text) and should be added in the footer
Re: Thank You for Your comment. We improved the text accordingly adding a footnote to the Table.
I would like to congratulate you on your interesting paper. I will be glad to review it one more time after revision.
Re: Thank You for Your comment. The manuscript has been significantly revised and improved according to the received suggestions.
Round 2
Reviewer 3 Report
The authors have addressed all my concerns adequately. Therefore, in my opinion, the paper can be accepted for publication. Nevertheless, some small mistakes (e.g. lack of a dot at the end of line 74) should be corrected in further steps.